# Priming by Insects: Differential Effects of Sympatric and Allopatric Priming upon Plant Performance and Tolerance to Herbivory

**DOI:** 10.3390/plants11243567

**Published:** 2022-12-17

**Authors:** Etzel Garrido, Karina Boege, César A. Domínguez, Juan Fornoni

**Affiliations:** Department of Evolutionary Ecology, Institute of Ecology, National Autonomous University of Mexico, Mexico City 04510, Mexico

**Keywords:** defence, local adaptation, plant–insect interactions, defensive strategy, tolerance

## Abstract

Plants have evolved multiple mechanisms to defend themselves from their multiple herbivores. Thus, being able to recognise among them and respond accordingly is fundamental for plant survival and reproduction. Defence priming prepares the plant to better or more rapidly respond to future damage; however, while it is considered an adaptive trait, to date, no studies have evaluated the extent and specificity of the priming recognition. To estimate the costs, benefits and specificity of priming, we used a highly specialist plant–insect system (*Datura stramonium*–*Lema daturaphila*) and performed a reciprocal transplant experiment with two populations where a priming stimulus (sympatric vs. allopatric) and a damage treatment (sympatric) were applied. We found no evidence of a fitness cost of priming, given that primed plants without damage showed no reduction in fitness. In contrast, our treatments affected the probability of bud abortion. That is, when damaged plants received no priming or the priming came from an allopatric insect, the likelihood of aborting the first bud was 1.9 times greater compared to plants being primed by their sympatric insect. We also found that damaged plants primed with an allopatric insect produced 14% fewer seeds compared to plants receiving a sympatric priming stimulus. Tolerance to herbivore damage was also the lowest when plants received the priming stimulus from an allopatric insect. Overall, these results suggest that, in our study system, plants recognise their local insect population reducing the negative effect of damage through a tolerance response.

## 1. Introduction

Since their appearance on Earth more than 400 million years ago, plants, as sessile organisms, have been evolving under biotic and abiotic stresses. Thus, being able to recognise different stressors and respond accordingly is fundamental for plant survival, development and reproduction. It is not surprising that plants have evolved numerous distinct sensing and signalling mechanisms to recognise potential harmful stresses [1]. Losing plant tissue is one of the main stresses faced by plants, and the evolution of multiple defence mechanisms support the strong selective pressure imposed by consumers [2]. Defence priming against herbivory involves a physiological state that allows plants to either better or more rapidly respond to future damage [1,3,4]. Since priming initiates a state of defence readiness once an attack occurs, one presumed benefit of priming is that it reduces or eliminates the costs associated with full implementation of an induced defence response. Defence priming is considered an adaptive, low-cost defensive strategy, given that defence responses are only activated and not fully expressed [5,6]. Hence, defence responses are deployed in a faster, stronger and/or more sustained manner following the perception of a subsequent challenging signal (the triggering stimuli) [7]. However, priming itself may result in potential costs because it requires changes in a regulatory network that are kept in a vigilant state until activated by a triggering stress [4]. Although the importance of defence priming as an adaptive trait is well-established, there are few, if any, attempts to examine potential costs.

Studying plant defence memory requires two steps—first, a priming event, which primes the defence-related response and second, a triggering stress, which activates the defence-related traits [8]. If priming provides fitness benefits, primed plants should express a more rapid and efficient defence response-compared to non-primed plants. This involves not only recognising cellular damage by chewing insects, for example, but the specific consumer species to activate the proper defensive strategy [9]. Ideally, the presence of defence priming could be assessed with a phenotypic analysis of the defensive state of a plant before and after a challenge with a biotic stress, combined with an assessment of the resulting cost–benefit balance of the induced response. It is also important to determine which plant-response variables are the most appropriate for evaluating the benefit of priming. Although this is well appreciated, surprisingly few studies have measured the fitness effects of defence priming [10].

Given that the benefit of the different defensive strategies (resistance or tolerance) depends on the coevolutionary state of the interaction [11], the fitness consequences of priming will depend on the extent of specificity in priming. Previous studies have shown that a plant’s compensation to herbivory is specific to herbivory species, type and intensities [12,13], but the extent of this specificity remains unknown. Additionally, most studies have focused on priming by HIPVs (Herbivore Induced Plant Volatiles; reviewed in [3]) and its consequences in plant resistant traits, but we still don’t know whether tolerance to herbivory can be elicited by a priming stimulus. If the coevolutionary process promotes a pattern of local adaptation, then plants should be able to recognise the most frequent consumer genotypes corresponding to their native habitat. In a previous experiment aimed at evaluating geographic variation in the degree of local co-adaptation, we found that damage by insects from a specific population (Tula, Hidalgo, Mexico, 20.05° N, 99.35° W) induced a greater tolerance response in plants from all other examined populations [11]. This particular result indicates that tolerance levels are influenced not only by the amount of damage that plants experience, but also by the provenance of the herbivore exerting the damage. In this work, we further examined the influence of the herbivore’s population origin in the defensive memory response after a priming stimulus. Using plants and insects from two populations, we performed a reciprocal transplant experiment where a priming stimulus of herbivores from two different populations (sympatric vs. allopatric) and a damage treatment (sympatric) were applied. The goals of the study were two-fold. First, we evaluated the benefit–cost balance of the priming stimulus and its possible changes given the provenance (origin) of the insect exerting it. Second, we evaluated the effects of the priming stimulus on plant performance and tolerance to herbivory.

## 2. Results

We found a significant main effect of plant population in the likelihood of aborting the first bud (χ^2^_1_ = 5.4713; *p* = 0.0193; Deviance (D^2^) = 0.2145) and seed production (*F*_1,107_ = 12.82; *p* = 0.0005), indicating population differentiation in these traits. Specifically, plants from Tula were 75% less likely to abort, which resulted in an increase in seed production (244.53 ± 20.24 seeds) relative to plants from Pedregal (164.20 ± 16.12 seeds). We also found differences in tolerance to herbivory (*F*_1,61_ = 3.94; *p* = 0.0516), albeit marginally, with plants from Pedregal being less tolerant (−3.24 ± 1.02) relative to plants from Tula (−1.19 ± 0.35). On the contrary, there were no differences in the root:shoot ratio (*F*_1,107_ = 0.23; *p* = 0.6336) and regrowth (*F*_1,107_ = 1.79; *p* = 0.1835) between both populations.

Interestingly, our treatments affected whether the plants aborted their first bud or not (χ^2^_4_ = 23.8046; *p* < 0.0001; D^2^ = 0.2145), their regrowth capacity (*F*_4,107_ = 4.47; *p* = 0.0022), seed production (*F*_4,107_ = 9.41; *p* < 0.0001) and tolerance to herbivory (*F*_2,61_ = 3.91; *p* = 0.0253). Control plants had a fifty percent chance of abortion; however, when the plants experienced herbivore damage without previous priming, or the priming came from an insect from an allopatric population, the likelihood of aborting the first bud was 1.9 times greater (Figure 1). Herbivore damage had a positive effect on regrowth (a possible tolerance mechanism), but priming did not change this effect (Figure 2). There were no differences in seed production between control plants and plants that only received the priming stimulus, suggesting that, at least in terms of seed production, there are no costs associated with being primed (Figure 3). We found that priming by sympatric insects did not prevent the reduction of seed production after herbivore damage (Figure 3). Moreover, when plants were primed by an allopatric insect, they produced even fewer seeds after being damaged (Figure 3), suggesting they were not able to recognise the stimulus given by an insect from another population. This same effect was also found in terms of the tolerance capacity to herbivory, given that plants primed by allopatric insects were less tolerant (Figure 4). Finally, there were no significant interactions between plant population and treatment for any variable, indicating that plants from both localities responded in the same way to our treatments.

## 3. Discussion

Our results show a significant insect-priming provenance effect on the likelihood of bud abortion, regrowth, seed production and tolerance to herbivory, indicating that allopatric vs. sympatric consumers provide differential information to their host plant. No apparent costs of priming were detected, supporting the expectation of a low-cost signalling strategy before severe damage. Different types of cues have been shown to prime a plant for improved defences, from leaf volatiles, through egg deposition and oral animal secretions to even colonisation with beneficial microorganisms (reviewed in [14]). In our experiment, the priming stimulus was performed by placing a larva on the plant, allowing it to eat a small amount of foliar tissue. Thus, our priming stimulus might include oral secretions [15], the release of regurgitant on the plant tissue [16] and insect-associated microbes [17]. Although we used the same insect species, it is still possible that there is population differentiation in the chemical and microbial composition of its oral secretions and regurgitant. Hence, future studies should consider the three-way interaction between microorganisms, insects and plants in the study of defence priming.

It has been proposed that defence priming includes all physiological, molecular and epigenetic changes that occur within the plant after the sensing of the stimulus (reviewed in [14]), which ultimately affects plant metabolism. Indeed, it has been shown that defence priming is regulated by different metabolic pathways [18]. For example, *Plantago lanceolata* increased primary metabolism to cope with aphids, but in the case of a pathogen attack, it increased systemic resistance [19]. Other studies have found effects related to primary metabolism only. In *Solanum lycopersicum*, priming with *Manduca sexta* regurgitant increased regrowth ability after intense mechanical defoliation [20], and in the grass *Leymus chinensis*, sheep saliva significantly increased the number of buds and biomass [21]. Although changes in resource allocation triggered by priming is a possible mechanism behind an increase in regrowth, there is still no evidence to support this hypothesis. Previous studies have shown that after damage, there is an increase in the transport of recently fixed carbon to roots and stems [22,23] and a reallocation of carbon to the roots, which leads to a depletion of carbohydrates [24]. Thus, understanding the relationship between herbivores’ mediated-priming on regrowth capacity and resource reallocation might help us understand the mechanisms that contribute to tolerance responses.

Theoretically, the priming state does not change a plant’s metabolism or gene expression until damage is experienced [18]. Thus, no or minimal fitness costs, in terms of growth and fruit or seed production, associated with priming are expected [1,5,6]. Here, we found no evidence of a fitness cost of priming, given that seed production was the same in control plants and plants receiving only the priming stimulus (Figure 3). Indeed, it has been shown that priming of *Arabidopsis thaliana* with β-aminobutyric acid (BABA) had no negative impact on seed production, although a minor reduction in plant growth rate was detected [6]. However, to our knowledge, there are still few published studies exploring the costs of priming in the context of plant–herbivore interactions, particularly in terms of seed production.

One of our more interesting results is that tolerance to herbivory decreased when the priming stimulus came from an allopatric insect relative to when plants were primed by a sympatric insect (Figure 4). This result alone suggests high specificity in the plant–herbivore system because the plants either receive different information from the allopatric insect or misinterpret such information. Plants are able to perceive Herbivore Associated Molecule Patterns (HAMPs), which are involve in insect recognition and the specific triggering of a defence response [16,25]. Among the two classes of HAMPs, those related to specific elicitors, such as oral secretions, oviposition fluids or even plant compounds modified by the insect herbivore [25], might help us explain why plants appear to recognise the provenance of the insect exerting the priming stimulus. If there are differences in HAMPs among different insect populations remains unknown. In any case, the primed tolerance response and higher seed production when priming and damage was imposed by the native insect population indicate a fitness benefit. One key issue to be explored is how plants recognise the signals and cues of an herbivore attack (see [26]), given that the adaptive benefit of priming should only be realised when a signal is a reliable indicator of a future stress. The interchange of information during chewing through the insect secretion and the associated holobiome is a new avenue to further understand the reciprocal evolutionary responses between plants and herbivores. To date, most of the studies about specificity of primed defences have focused on understanding resistance mechanisms. It has even been shown that induced resistance is specific enough, so that feeding by two closely related species of whiteflies induces salicylic acid defences, while suppressing jasmonic acid ones [27]. Moreover, there is evidence of plants’ perception of specific volatiles and specificity of elicitation of primed resistance in maize [28] and native tobacco [29]. In contrast, few studies have shown that tolerance responses to herbivory are specific to the intensity and identity of herbivory [12,13]. Our results show that both plant populations responded similarly to the allopatric vs. sympatric priming stimuli, indicating a generalised response of tolerance to native consumers. Local adaptation of plants also involves recognising clues about their specific consumer that goes behind the occurrence of damage. Specifically, Gavloski and Lamb [13] found that compensation was correlated with a balanced root:shoot ratio [30]. Here, we also estimated changes in the root:shoot ratio; however, we found that neither priming nor herbivory affected this variable. Moreover, plant regrowth did not change given the priming stimulus; thus, our results suggest that even an equal recovery of foliage after damage does not always assure a complete recovery of plant fitness.

## 4. Materials and Methods

### 4.1. Study System

*Datura stramonium* L. (Solanaceae) is the main host of the specialist leaf beetle *Lema daturaphila* Kogan & Goeden (Coleoptera: Chrysomelidae). All but the pupal stage occur on the leaf tissue of its host, where it can survive and reproduce for up to four generations per season (E. Garrido, *pers. obs.*). Overall, plants of *D. stramonium* experience levels of damage between 10–50% of total leaf area, reducing their seed production [31]. While plant resistance has a negative effect on the survival of the insect, tolerance does not [32]. In a previous experiment aimed at evaluating co-adaptation levels among four populations in Central Mexico, we found geographic variation in the degree of reciprocal local adaptation [11]. Herbivores from the locality of Pedregal were locally adapted; that is, insects achieved higher overall performance when eating plants from Pedregal compared to herbivores eating plants from other localities. On the other hand, herbivores from the Tula locality showed no evidence of local adaptation, given that their performance when eating plants from Tula was similar to their performance when eating plants from other localities [11]. Surprisingly, when plants from all populations were consumed by herbivores from Tula, we found an increase in tolerance to herbivory, suggesting that insects from Tula might trigger tolerance responses in the plants they consume. Due to these previous results, seeds and insects from Pedregal (19.32° N, 99.19° W) and Tula (20.05° N, 99.35° W) were used for this study.

### 4.2. Experimental Design

During the summer of 2011, seeds from 30 maternal families per population were germinated. Two weeks after germination, 60 plants per population (*n* = 120) were individually transplanted into 4 L pots, filled with potting soil and placed in a glasshouse at the Instituto de Ecología (UNAM). Simultaneously, c. 100 adults of *L. daturaphila* from both populations were collected and taken to the laboratory. We followed a full-factorial experimental design where naive, primed and triggered plants were manipulated. The experiment consisted of the following five treatments: control plants (−Ps, −H); primed plants without herbivore damage (+Ps, −H); naive plants with herbivore damage (−Ps, +H); plants primed with their sympatric insect and later exposed to herbivore damage (+Ps, +H); and plants primed with an allopatric insect and later exposed to herbivore damage (+Pa, +H). Priming in sympatry (Ps) implies that plants from Pedregal were primed by insects from Pedregal and plants from Tula were primed by insects from Tula. Priming in allopatry (Pa) implies that plants from Pedregal were primed by insects from Tula and vice versa. This design allowed us to evaluate the costs, benefits and specificity of priming as a function of plant and insect provenance. The priming stimulus was applied when the fifth leaf fully expanded (two months after germination, approximately) and consisted of placing one third-instar larva on each of leaves five, six and seven (*n* = three larvae per plant) and allowing them to consume about one cm^2^ of foliar tissue per leaf. One week after priming, an herbivore damage treatment was applied, where five third-instar larvae were placed on each of the leaves of the plants until 50% of the leaf area was damaged. Since consumption differed among larvae and some died, new larvae were added as needed until approximately 50% of the leaf tissue was consumed, which took between five and seven days to occur. Afterwards, the remaining larvae were removed from all the experimental plants.

### 4.3. Response Variables

During the experiment, we recorded whether the plants aborted their first bud. At the end of the experiment, when most of the fruits were almost ripe, all the plants were harvested. The length of leaves produced after herbivore damage was recorded to later estimate total new leaf area per plant. The relationship between leaf length and leaf area (leaf area = 0.329 * leaf length^2^; R^2^ = 0.98, [33]) was used to estimate new leaf area per plant and was considered as an estimation of plant regrowth. The shoots and roots were separately collected, dried for two days in an oven at 35 °C and weighed. The root:shoot ratio was then calculated. The total number of seeds per plant was also counted. Finally, tolerance to herbivory was estimated as follows: tolerance = (D − U)/D, where D and U stand for damaged and undamaged plants, respectively (see [34]). Therefore, tolerance values represent the proportional increase/decrease in seeds as a result of herbivore damage and include both genetic and environmental effects.

### 4.4. Statistical Analyses

All the analyses and plots were performed using R [35] and the ggplot2 library [36]. All the variables except bud abortion were analysed with a two-way ANOVA, including plant population, treatment, their interaction and block as sources of variation. The likelihood of aborting the first bud was analysed using a GLM with binomial distribution and “logit” link function.

## Figures and Tables

**Figure 1 plants-11-03567-f001:**
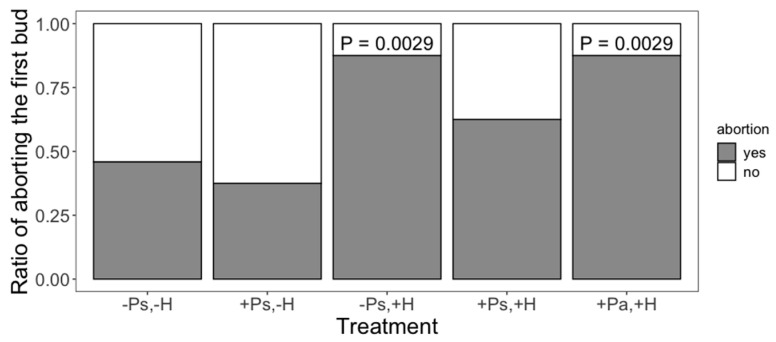
Effect of both priming (P) and herbivory (H) on the likelihood of aborting the first bud. Plants were more likely to abort when damaged if they received no priming or the priming came from an allopatric insect. Plus (+) and minus (−) signs denote the presence/absence of the treatment; Ps and Pa denote priming being performed by a sympatric or allopatric insect, respectively. There were 24 replicates (plants) per treatment (*n* = 120).

**Figure 2 plants-11-03567-f002:**
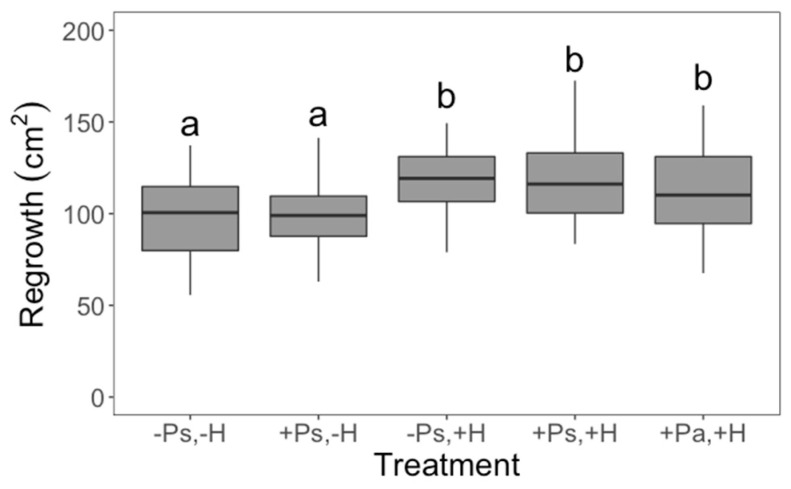
Effect of herbivory (H) on plant regrowth. Boxplots representing the median, minimum and maximum values are shown. Plants that experienced 50% of damage increased their leaf area relative to undamaged plants. Plus (+) and minus (−) signs denote the presence/absence of the treatment; Ps and Pa denote priming being performed by a sympatric or allopatric insect, respectively. Different letters denote significant differences following a Tukey–Kramer test (*p* < 0.05). There were 24 replicates (plants) per treatment (*n* = 120).

**Figure 3 plants-11-03567-f003:**
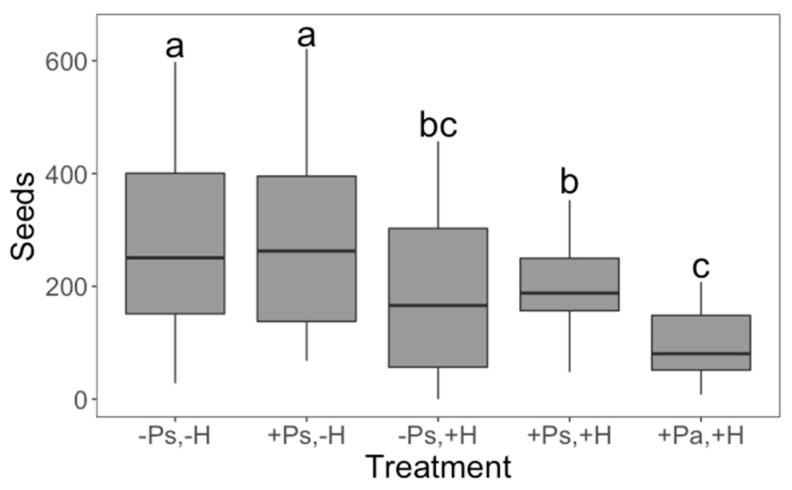
Effect of both priming (P) and herbivory (H) on seed production. Boxplots representing the median, minimum and maximum values are shown. Plants that experienced 50% of damage produced fewer seeds relative to undamaged plants. However, when primed with an allopatric insect, the negative effect of damage was higher. Plus (+) and minus (−) signs denote the presence/absence of the treatment; Ps and Pa denote priming being performed by a sympatric or allopatric insect, respectively. Different letters denote significant differences following a Tukey–Kramer test (*p* < 0.05). There were 24 replicates (plants) per treatment (*n* = 120).

**Figure 4 plants-11-03567-f004:**
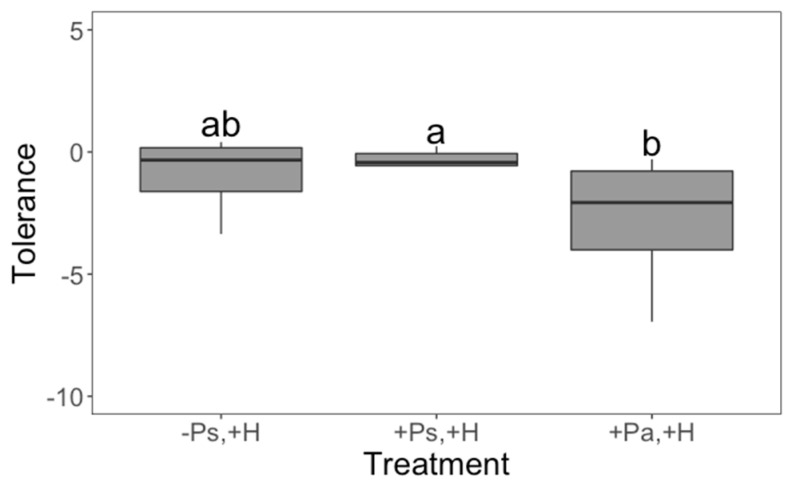
Effect of both priming (P) and herbivory (H) on tolerance. Boxplots representing the median, minimum and maximum values are shown. When plants were primed by an allopatric insect, the tolerance capacity was lower. Plus (+) and minus (−) signs denote the presence/absence of the treatment; Ps and Pa denote priming being performed by a sympatric or allopatric insect, respectively. Different letters denote significant differences following a Tukey–Kramer test (*p* < 0.05). There were 24 replicates (plants) per treatment (*n* = 72).

## Data Availability

Not applicable.

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
