# Peer review of "Priming by Insects: Differential Effects of Sympatric and Allopatric Priming upon Plant Performance and Tolerance to Herbivory"

_plants, 2022, doi:10.3390/plants11243567_

Round 1

Reviewer 1 Report

This paper investigates the effects of distinct insect populations on plant performance and tolerance. I have only a few comments/queries.

Define sympatric and allopatric in the context of the experiments described in this paper.

Make it clear in the figure legends how many times the experiments were repeated to give the n value for statistical analysis. In figure 1 how often was the treatment repeated? With x repeats, is it possible to provide a P value?

page 2 Materials/Methods: 2.1: explain what is meant by local adaptation in this context, that is, provide some detail from the results in reference 11. 

page 4 para 2 line 7: please explain, as herbivory appears to increase regrowth in figure 2. 

Perhaps the authors could discuss more on the role of defence genes in plants, either in the introduction or discussion.

Author Response

This paper investigates the effects of distinct insect populations on plant performance and tolerance. I have only a few comments/queries.

  1. Define sympatric and allopatric in the context of the experiments described in this paper.

Response: We thank the reviewer for noticing that a better explanation of what we mean by sympatric and allopatric treatments improves the manuscript. Please see lines 275-277.

  1. Make it clear in the figure legends how many times the experiments were repeated to give the n value for statistical analysis. In figure 1 how often was the treatment repeated? With x repeats, is it possible to provide a P value?

Response: We agree with the reviewer in that figure legends would be clearer if more information is added. Specifically, in Figure 1 we have changed the asterisks for the exact P-values and made the changes in the legend accordingly.

  1. Page 2 Materials/Methods: 2.1: explain what is meant by local adaptation in this context, that is, provide some detail from the results in reference 11.

Response: We have added this information. Please see lines 253-257.

  1. Page 4 para 2 line 7: please explain, as herbivory appears to increase regrowth in figure 2.

Response: We thank the reviewer for noticing this inconsistency. We have made the changes accordingly. Please see lines 115-116.

  1. Perhaps the authors could discuss more on the role of defence genes in plants, either in the introduction or discussion.

Response: We thank the reviewer for this suggestion. Although the topic of the present manuscript does not relate to gene expression, we have added some information about how herbivore associated molecule patterns (HAMPs) might help us explain the different tolerant responses when primed by allopatric vs sympatric insects (see lines 213-219) which ultimately implies changes in gene expression.

Reviewer 2 Report

I presume a ‘Communication’ is similar to a ‘Scientific Note’ since there appears to be only one set of experiments reported in the manuscript.  If a ‘Communication’ is essential a preliminary style report, this research should be published as such.  My primary question is this research appeared to be conducted in 2011, why is it only now being reported?  Also, was there follow-up research that has been reported?

Specific comments and suggestions are included in the annotated manuscript.

Author Response

I presume a ‘Communication’ is similar to a ‘Scientific Note’ since there appears to be only one set of experiments reported in the manuscript. If a ‘Communication’ is essential a preliminary style report, this research should be published as such.

  1. My primary question is this research appeared to be conducted in 2011, why is it only now being reported? Also, was there follow-up research that has been reported?

Response: Yes, the research reported here was performed during the summer of 2011. We have no specific reason for why we just finished writing this manuscript. After collecting the data, the first author spent five years as a postdoc in other labs and then started her own group. It was until we were asked to make a contribution to the Special Issue that we decided to finally finished this manuscript. And no, we have not reported any follow-up research on this particular matter.

  1. Specific comments and suggestions are included in the annotated manuscript.

Response: We really appreciate all the comments and suggestions made. We have included all of them in the revised version of the manuscript.

** The following comments were given in the annotated version

  • What is the D-statistic, isn't it used in multi-variate analyses?

Response: The D2 represent the deviance of the model. We analysed the probability of bud abortion using a Generalised Linear Model (glm) with binomial distribution because the response variable is whether the plants aborted or not (two possible results). It is common practice to report the D2 value when performing glm´s.

- It would be interesting to see how many generations of exposure it would take for priming by an allopatric population to produce results similar to a sympatric population.

Response: We totally agree with this comment. We will consider doing a follow-up experiment to address this question.